# Health system costs and days in hospital for colorectal cancer patients in New South Wales, Australia

David E. Goldsbury[1,2]*, Eleonora Feletto[1], Marianne F. Weber[1], Philip Haywood[2,3], Alison Pearce[2], Jie-Bin Lew[1], Joachim Worthington[1], Emily He[1,4], Julia Steinberg[1], Dianne L. O'Connell[1,5], Karen Canfell[1,6]

1 The Daffodil Centre, The University of Sydney, A Joint Venture with Cancer Council NSW, Sydney, NSW, Australia, 2 School of Public Health, University of Sydney, Sydney, NSW, Australia, 3 Centre for Health Economics Research and Evaluation, University of Technology Sydney, Sydney, NSW, Australia, 4 Gastroenterology and Liver Department, Concord Hospital, Sydney, NSW, Australia, 5 School of Medicine and Public Health, University of Newcastle, Newcastle, NSW, Australia, 6 Prince of Wales Clinical School, University of New South Wales, Sydney, NSW, Australia

* davidg@nswcc.org.au

## Abstract

### Introduction

Colorectal cancer (CRC) care costs the Australian healthcare system more than any other cancer. We estimated costs and days in hospital for CRC cases, stratified by site (colon/rectal cancer) and disease stage, to inform detailed analyses of CRC-related healthcare.

### Methods

Incident CRC patients were identified using the Australian 45 and Up Study cohort linked with cancer registry records. We analysed linked hospital admission records, emergency department records, and reimbursement records for government-subsidised medical services and prescription medicines. Cases' health system costs (2020 Australian dollars) and hospital days were compared with those for cancer-free controls (matched by age, sex, geography, smoking) to estimate excess resources by phase of care, analysed by sociodemographic, health, and disease characteristics.

### Results

1200 colon and 546 rectal cancer cases were diagnosed 2006–2013, and followed up to June 2016. Eighty-nine percent of cases had surgery, chemotherapy or radiotherapy, and excess costs were predominantly for hospitalisations. Initial phase (12 months post-diagnosis) mean excess health system costs were $50,434 for colon and $60,877 for rectal cancer cases, with means of 16 and 18.5 excess hospital days, respectively. The annual continuing mean excess costs were $6,779 (colon) and $8,336 (rectal), with a mean of 2 excess hospital days each. Resources utilised (costs and days) in these phases increased with more advanced disease, comorbidities, and younger age. Mean excess costs in the year before death were $74,952 (colon) and $67,733 (rectal), with means of 34 and 30 excess hospital

**Data Availability Statement:** The data cannot be made available by the authors. These are third party data not owned or collected by the authors and on-provision is not permitted by the relevant

data custodians (Sax Institute, Services Australia, NSW Ministry of Health), as it would compromise the participants' confidentiality and privacy. The data contain potentially identifying and sensitive patient information. However the data are available from the data custodians for approved research projects. Data access enquiries can be made to the Sax Institute (https://www.saxinstitute.org.au/our-work/45-upstudy/governance/). Other researchers would be able to access these data using the same process followed by the authors.

**Funding:** DEG, EF, MFW, PH, JS and KC are investigators on a study of cancer patient population projections, funded by the Australian Government Medical Research Future Fund (MRFF) – Preventive and Public Health Research Initiative – 2019 Targeted Health System and Community Organization Research Grant Opportunity (MRF1200535). No funder had a role in study design, analysis, decision to publish, or preparation of the manuscript. KC and JBL received grants from the National Health and Medical Research Council of Australia (APP1194679 and APP1194784, respectively) during the conduct of the study. KC is an investigator on an unrelated investigator-initiated trial of cytology and primary HPV screening in Australia (Compass), which is conducted and funded by the Victorian Cytology Service (VCS), a government-funded health promotion charity. The VCS has received equipment and a funding contribution for the Compass trial from Roche Molecular Systems and Ventana. However, neither the authors nor the authors' organisations receive direct funding from industry for this trial or any other project.

**Competing interests:** The authors have declared that no competing interests exist.

days, respectively–resources utilised were similar across all characteristics, apart from lower costs for cases aged ≥75 at diagnosis.

## Conclusions

Health system costs and hospital utilisation for CRC care are greater for people with more advanced disease. These findings provide a benchmark, and will help inform future cost-effectiveness analyses of potential approaches to CRC screening and treatment.

## Introduction

Colorectal cancer (CRC) has the third highest incidence of all cancers worldwide, with over 1.9 million new cases in 2020 [1]. There are over 15,000 new cases each year in Australia, where CRC is the second most common cause of cancer death [2]. CRC often requires intensive treatment and hospitalisation shortly after diagnosis, both of which place a substantial burden on the healthcare system [3, 4]. CRC care costs the Australian healthcare system more than any other cancer, with health system costs estimated at over $1 billion Australian dollars (AUD) in 2013 [5]. Guideline-recommended treatments are different for colon and rectal cancers, and vary by disease stage, resulting in differences in the associated resource utilisation [4]. However, there is limited information about per-person resource utilisation (healthcare costs or numbers of days in hospital) for colon and rectal cancers by disease stage in Australia, and how the specific healthcare components contribute to costs.

Australia has a government-funded universal healthcare system for all permanent residents, along with a private healthcare system largely paid for by patients and health insurance providers. There is also a government-funded National Bowel Cancer Screening Program (NBCSP), which aims to reduce the morbidity and mortality from CRC [6, 7]. The NBCSP sends an immunochemical faecal occult blood test (FOBT) to Australians aged 50–74 every two years. The program began in 2006 and reached full implementation in 2020, with a participation rate of 44% in 2018–2019 [8]. Evaluations of the current NBCSP based on available data have shown this current screening approach is highly cost-effective, largely by enabling treatment of precancerous lesions and/or enabling cancer diagnoses at earlier stages where treatment options can be more effective [9–11]. Screening programs can be continuously improved, but evaluations require contemporary data on costs and patient management. In the absence of other information, our prior modelled cost-effectiveness and impact evaluations of NBCSP [9–13] have utilised a 2011 study reporting CRC treatment costs from a hospital-based setting [14].

A more nuanced characterisation of CRC costs and the way in which they are incurred in Australia is crucial for further economic evaluations of CRC prevention and control strategies. This will inform future analyses of CRC-related healthcare needs, including cost-effectiveness analyses of new approaches to CRC screening and treatment. Past studies have considered costs incurred by specific treatments [14–16], and our previous work assessed total excess costs to capture the overall burden on resources including services and/or care not flagged as being specific to cancer (e.g. extra GP/specialist consultations) [5]. The latter can be more informative when evaluating total cost-savings from prevention and screening strategies. Our initial work on the costs of cancer in Australia estimated excess costs for all cancers combined, and for the most common cancer types [5], but did not estimate how costs differ for colon and rectal cancers, by stage at diagnosis, and by patients' characteristics (e.g. age, health insurance

status). These more detailed estimates of healthcare costs are required to inform economic evaluations of new treatments specific to cancer site or stage, as well as prevention or screening approaches targeted to population subgroups based on age and/or other characteristics. Similarly, estimates for hospital inpatient days are needed to determine the potential impact that changes in CRC control could have on hospital resource requirements.

In the current study we aimed to estimate health system costs and days spent in hospital by disease stage at diagnosis for individuals with colon or rectal cancer in New South Wales (NSW), Australia's most populous state. We used detailed, person-level information to generate population-based estimates, analysing administrative health records linked with self-reported data from a large prospective cohort study of NSW residents. Costs and hospital days were estimated by phase of care and year/month around diagnosis and death, compared with those for matched cancer-free controls and disaggregated into the types of health services utilised. We also aimed to disaggregate the estimates using several characteristics potentially related to health services use and/or CRC incidence, such as age, sex, health insurance status and accessibility of services.

## Materials and methods

### Source data

The source population was all 267,153 participants in the Sax Institute's NSW 45 and Up Study. A detailed description of the study cohort has been provided previously [17]; in brief, potential study participants were sampled from the Medicare enrolment database held by Services Australia (formerly the Department of Human Services), which provides near-complete coverage of the Australian population. People aged >80 years and those in rural areas were oversampled. Participants completed a paper-based health and lifestyle questionnaire during 2006–2009 ("baseline"), and consented to have their questionnaire data linked with their health-related records held in routinely collected, administrative datasets.

The linked health records included reimbursements for government-subsidised prescription medicines in the Pharmaceutical Benefits Scheme (PBS), and outpatient and medical services (e.g. GP/specialist consultations, colonoscopies, CT scans, pathology tests), and some in-hospital procedures subsidised through the Medicare Benefits Schedule (MBS). The Sax Institute linked these records for study participants using a unique identifier. The Centre for Health Record Linkage [18] probabilistically linked the other health records for study participants, using data provided by the NSW Ministry of Health, including inpatient care in all NSW hospitals recorded in the Admitted Patient Data Collection (APDC), emergency presentations in the Emergency Department Data Collection (EDDC), statutory cancer notifications (not keratinocyte/non-melanoma skin cancers) in the NSW Cancer Registry (NSWCR, from Cancer Institute NSW), and death notifications in the NSW Registry of Births, Deaths and Marriages (RBDM) (Fig 1).

### Study sample

We excluded participants with (see Fig 2):

i.  probable linkage errors and those aged <45 years at baseline.

ii.  a NSWCR record of another invasive cancer prior to the incident CRC.

iii.  self-reported history of cancer at baseline (except keratinocyte cancers).

iv.  healthcare subsidised by the Australian Government Department of Veterans' Affairs (DVA), as their prescription medicines are recorded in a different billing system not

**Fig 1. Data sources and date coverage.** NSWCR: New South Wales Cancer Registry; MBS: Medicare Benefits Schedule; PBS: Pharmaceutical Benefits Scheme; RBDM: Registry of Births, Deaths and Marriages. Reproduced from reference [5].

available for this study. DVA clients were identified via self-report or any mention of DVA coverage in hospital or ED records.

v. unknown day/month of diagnosis in NSWCR, or CRC first recorded on their death certificate.

We identified cases as participants with a NSW Cancer Registry record of incident CRC after recruitment, up until December 2013. Cases were identified using topography codes C18 (colon) and C19-C20 (rectal) from the International Statistical Classification of Diseases and Related Health Problems, Tenth Revision (ICD-10).

Cases were matched to controls who had no NSWCR or self-reported record of cancer at any time and who were alive at the diagnosis date of their matched case. Up to four controls were matched to each case by age (±5 years), sex (female; male), Local Government Area of residence (153 areas in NSW) and smoking status (never smoker; ex-smoker quit >15 years; ex-smoker quit ≤15 years; current smoker). This is an extension of the matching used in previous studies that estimated excess costs by age, sex and some geographical areas [19–21]. Information for matching was ascertained from baseline data. People with missing responses for any of the matching variables were excluded unless the information could be imputed from other data sources.

## Costs and hospital days

Direct health system costs (healthcare payer perspective) from the MBS and PBS databases were identified from the costs listed in individual claims records. Inpatient hospital costs were derived from APDC records by linking the Australian Refined Diagnostic Related Group (AR-DRG) code for each hospitalisation to the average public hospital admission cost recorded in the 2010 National Hospital Cost Data Collection [22]. From July 2015 onwards, private hospital data did not include AR-DRG codes (7% of all admissions in this study), but 8 in 10 of these hospitalisations could be assigned AR-DRGs based on the codes used for the same admission types in records from before 2015. This left 1% of admissions without an AR-DRG–

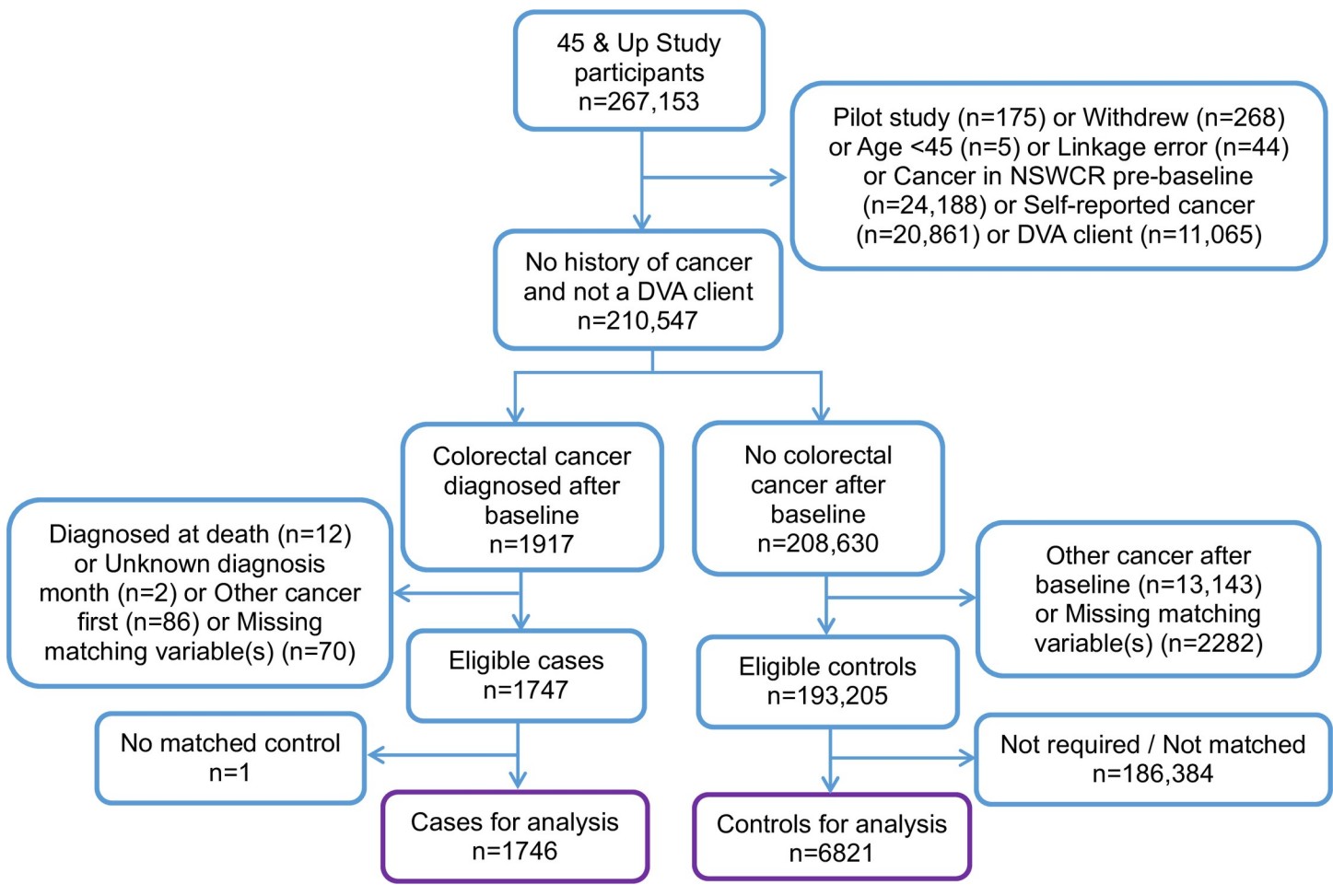

**Fig 2. Cohort selection flow diagram.**

almost all >2 years after diagnosis–and these were assigned a cost of $1500 for the first day and $300 per additional day. To put these assumed costs into perspective, single-day hospitalisations costs were ~$2200 for a colonoscopy and ~$1000 for rehabilitation. A sensitivity test using double these assigned costs made <1% difference to the overall results. ED presentations were assigned average costs by triage category and discharge status from the National Hospital Cost Data Collection [22]. APDC and EDDC costs were combined to give hospital-based costs. All cost values were converted to 2020 Australian dollars using the Australian Health Index [23]. The number of calendar days spent in hospital was calculated using APDC hospital admission records. A small proportion of people (1%) had more than two hospitalisations on a single day–these were counted as a single day in hospital.

Costs and hospital days were estimated for three phases of care: initial, continuing, and terminal. The phases for each case-control group were assigned based on the case's diagnosis date, death date and the end of follow-up as follows:

- For cases who died before July 2016, the final year up to death was designated the terminal phase. If the case died ≤1 year after diagnosis, the terminal phase started at the diagnosis date and no costs/hospital days were attributed to other phases.

- For cases who survived >1 year but ≤2 years, the initial phase was the period from diagnosis until the start of the 12-month terminal phase.

- For cases who survived at least two years, the first year after diagnosis was designated the initial phase, and the continuing phase was the period between the end of the initial phase and the start of the terminal phase or 30 June 2016. For cases who died during July 2016—June 2017, the continuing phase ended one year prior to their death. Costs and hospital days for the continuing phase were annualised, and we did not include case-control groups with a continuing phase of <3 months (2% of groups).

We also estimated the annual costs and hospital days for each 12-month period around the case's diagnosis date, from 2 years pre-diagnosis to 5 years post-diagnosis, along with monthly costs and hospital days from 3 months pre-diagnosis to 6 months post-diagnosis. For cases who died, we estimated the monthly excess end-of-life costs and hospital days for the last 6 months relative to their death date. The results for each of these time periods are presented to allow for use in future modelling studies, and to highlight the key time periods for resource use.

## Statistical analyses

The excess costs and hospital days due to CRC were estimated for each case by taking their total costs and hospital days and subtracting the average costs and days for their matched controls. If the case was still alive and any of their matched controls died, the case and their remaining matched control(s) were included in any subsequent calculations, or they were censored if there were no remaining matched controls (Fig A in S1 File). If the case died, then the included costs and days were censored at that date for annual and monthly calculations, and the case-control group was excluded from analyses for subsequent time periods. Means of excess costs and hospital days were estimated, along with the proportions of excess costs contributed by each source (inpatient hospitalisations and ED presentations; prescription medicines in the PBS; services in the MBS). Costs were assigned to the relevant time period (phase/month/year relative to diagnosis/death) using the date of the MBS service or PBS supply or ED presentation. Hospitalisations could span multiple time periods, such as straddling the first and second year after diagnosis, or the initial phase and continuing phase. When this occurred, hospitalisation costs were apportioned across time periods using the proportion of days in each time period out of the total length of stay for that hospitalisation. Medians, standard deviations, and inter-quartile ranges (IQRs) are reported in the (S1 File).

Cases were classified by cancer site (colon, rectal), NSWCR summary spread of disease at diagnosis ("stage": localised; regional; distant metastases; unknown–based on notifications received by the NSWCR up to 4 months after diagnosis), sex, age at diagnosis (45–54 years; 55–64; 65–74; ≥75), year of diagnosis, smoking status at baseline, remoteness of place of residence at baseline [24], quintile of socioeconomic disadvantage of place of residence at baseline [25], health insurance status at baseline (private health insurance; healthcare concession card; none), body mass index (BMI: normal/underweight ($<25kg/m^2$), overweight (25 to <30), obese (≥30)), comorbidities based on a modified Charlson Comorbidity Index [26] using hospital admissions up to 5 years prior to diagnosis (scores 0, 1, ≥2), and baseline self-reported CRC screening (any screening, and separately for FOBT). These characteristics were included due to their association with treatment and survival [27, 28]. Where possible, missing values were imputed from information for the same person recorded in other datasets. All excess costs and days were analysed using weights so that the stage distribution in our study matched the distribution of all colon/rectal cancers diagnosed in NSW during 2011–2015 [29].

We estimated excess costs and hospital days stratified by cancer site, with disease stage as the primary covariate of interest. We further tested the association between excess costs or days and all cases' characteristics described in the previous paragraph using a multivariable gamma regression with a log link, simultaneously adjusting for all other covariates, and excluding cases with a missing value for any of the covariates. Due to the number of missing values for BMI and self-reported screening, these were only included in the multivariable regressions if the corresponding p-value was <0.10. A small number of outlying values were excluded (standardised Pearson residual <-4 or >5). Due to potentially negative excess costs for individual cases, all cost estimates were "offset" by +$50,000 for this specific analysis so that all were >$0 and could be included in the gamma regression. Similarly, hospital days were offset by +50 days. A sensitivity analysis offset by +$100,000 and +100 days showed little difference in the results. Finally, we also estimated the proportions of cases having anti-cancer treatment (surgery, chemotherapy, radiotherapy), ascertained from APDC, MBS and PBS records for 30 days pre-diagnosis onwards (the corresponding codes/item numbers are listed in Table A in S1 File). For this analysis, we considered a record of supply of a government-subsidised medicine as having "had" the medicine.

Analyses were carried out using SAS v9.4 (SAS Institute Inc., NC, US) and Stata (StataCorp, College Station, TX, US). Ethical approval for the 45 and Up Study was provided by the University of NSW Human Research Ethics Committee (HREC/10186) and specific approval for this analysis was provided by the NSW Population and Health Services Research Ethics Committee (HREC/14/CIPHS/54).

## Results

There were 1747 eligible CRC cases: 95% (n = 1665) had four matched controls, 2% had three matched controls, 1% had two matched controls, 2% had one matched control and one case had no matched controls and was excluded. The final study sample comprised 1,200 colon cancer and 546 rectal cancer cases. The median age at diagnosis was 72 years for colon cases and 68 for rectal cases, matching the median age of all colon and rectal cancer cases in NSW, and the unweighted stage distributions by cancer site were similar to the state-wide distributions (Table B in S1 File). The proportion of cases in regional/remote areas was higher than for NSW overall, due to the oversampling in the 45 and Up Study. Eighty-nine percent of cases had a record of anti-cancer treatment (Table 1). Twelve percent of cases were treated with one of the reasonably well-established monoclonal antibodies (bevacizumab from 2009, cetuximab from 2010), but there were no records of subsidised use of the more recently introduced immunotherapies, as expected based on the time period covered in the available data. Sixty-five percent of all cases survived ≥5 years after diagnosis. Among cases with metastases at diagnosis, 50% of colon cases and 33% of rectal cases died within one year of diagnosis.

### Initial care phase

In the initial phase, the mean excess costs were $50,434 for colon cases (IQR $25,448-$61,574; see Table D in S1 File) and $60,877 for rectal cases (IQR $31,801-$82,246). Hospital-based care accounted for 79% and 75% of these health system costs, respectively (Table 2). Anti-cancer treatment was common: 76% of colon cancer cases and 71% of rectal cancer cases had surgery in the initial treatment phase, 29% and 45% respectively had chemotherapy (higher for regional/distant stage cases), and 1% and 28% respectively had radiotherapy (Table F in S1 File). The mean number of excess days in hospital in the initial phase was 16 days for colon cases (IQR 6–20; Table E in S1 File) and 18.5 days for rectal cases (IQR 7–25). The mean length of stay for surgery admissions was 11 days for colon cases (median 9, IQR 7–11), and 12 days

**Table 1. Sociodemographic, diagnosis and treatment characteristics of colon and rectal cancer cases diagnosed 2006–2013 in the 45 and Up Study.**

| | Colon cases (n = 1200) | | Rectal cases (n = 546) | |
|---|---|---|---|---|
| | No. of cases | % of cases | No. of cases | % of cases |
| Age at diagnosis (years) | | | | |
| *Median age (IQR)* | *72 (65–80)* | | *68 (61–76)* | |
| 45–54 | 62 | 5% | 58 | 11% |
| 55–64 | 220 | 18% | 147 | 27% |
| 65–74 | 417 | 35% | 183 | 34% |
| ≥75 | 501 | 42% | 158 | 29% |
| Sex | | | | |
| Female | 624 | 52% | 191 | 35% |
| Male | 576 | 48% | 355 | 65% |
| Remoteness of place of residence | | | | |
| Major cities | 630 | 53% | 256 | 47% |
| Inner regional | 413 | 34% | 210 | 38% |
| Outer regional/Remote/Very remote | 157 | 13% | 80 | 15% |
| Area-level socioeconomic quintile | | | | |
| Most disadvantaged quintile | 298 | 25% | 126 | 23% |
| Quintile 2 | 257 | 21% | 122 | 22% |
| Quintile 3 | 222 | 19% | 109 | 20% |
| Quintile 4 | 181 | 15% | 94 | 17% |
| Least disadvantaged quintile | 216 | 18% | 85 | 16% |
| *Missing* | *26* | *2%* | *10* | *2%* |
| Health insurance status at baseline | | | | |
| Private insurance | 691 | 58% | 315 | 58% |
| Concession card | 322 | 27% | 129 | 24% |
| None | 155 | 13% | 86 | 16% |
| *Missing* | *32* | *3%* | *16* | *3%* |
| Smoking status at baseline | | | | |
| Never smoker | 649 | 54% | 251 | 46% |
| Ex-smoker quit >15 years | 325 | 27% | 170 | 31% |
| Ex-smoker quit ≤15 years | 136 | 11% | 81 | 15% |
| Current smoker | 90 | 8% | 44 | 8% |
| Body Mass Index | | | | |
| Normal/Underweight ($<25kg/m^2$)[a] | 422 | 35% | 181 | 33% |
| Overweight (25-<30) | 436 | 36% | 209 | 38% |
| Obese (≥30) | 259 | 22% | 126 | 23% |
| *Missing* | *83* | *7%* | *30* | *5%* |
| Baseline screening information[b] | | | | |
| Ever had CRC screening | 542 | 45% | 160 | 29% |
| Ever had FOBT | 289 | 24% | 102 | 19% |
| Year of diagnosis | | | | |
| 2006–2008 | 165 | 14% | 93 | 17% |
| 2009 | 204 | 17% | 90 | 16% |
| 2010 | 219 | 18% | 103 | 19% |
| 2011 | 216 | 18% | 97 | 18% |
| 2012 | 187 | 16% | 73 | 13% |
| 2013 | 209 | 17% | 90 | 16% |

(*Continued*)

**Table 1.** (Continued)

| | Colon cases (n = 1200) | | Rectal cases (n = 546) | |
|---|---|---|---|---|
| | No. of cases | % of cases | No. of cases | % of cases |
| Stage at diagnosis | | | | |
| Localised | 398 | 33% | 187 | 34% |
| Regional | 492 | 41% | 221 | 40% |
| Distant metastases | 243 | 20% | 90 | 16% |
| Unknown | 67 | 6% | 48 | 9% |
| Charlson comorbidity score | | | | |
| 0 | 990 | 83% | 474 | 87% |
| 1 | 116 | 10% | 42 | 8% |
| ≥2 | 94 | 8% | 30 | 5% |
| Anti-cancer treatment | | | | |
| Surgery | 1014 | 85% | 429 | 79% |
| Chemotherapy | 469 | 39% | 302 | 55% |
| Radiotherapy | 90 | 8% | 213 | 39% |
| Any of the above | 1074 | 90% | 487 | 89% |
| Survival time | | | | |
| ≤1 year | 189 | 16% | 62 | 11% |
| >1–2 years | 91 | 8% | 33 | 6% |
| >2–3 years | 56 | 5% | 35 | 6% |
| >3 years | 864 | 72% | 416 | 76% |
| *>5 years[c]* | *566/880* | *64%* | *281/420* | *67%* |

[a] "Underweight" (<18.5) accounted for ~1% of colon and rectal cancer cases.

[b] The question asked about ever being screened or having FOBT and was not specific to the National Bowel Cancer Screening Program.

[c] For cases diagnosed up to June 2012 (death records up to June 2017 were available).

CRC: colorectal cancer; FOBT: faecal occult blood test; IQR: inter-quartile range.

for rectal cases (median 10, IQR 7–14), with little difference in medians for public/private hospitals but some longer admissions in public hospitals.

Excess costs and hospital days were substantially higher for those with distant metastases or regional disease at diagnosis (Tables 2 and 3). After adjusting for all key covariates, higher costs for colon cancer cases in the initial phase were associated with metastatic or regional disease, higher comorbidity score and diagnosis at age <55, with lower costs for cases aged ≥75, current smokers, cases with no health insurance and those from inner regional and rural areas (Fig 3; Table G in S1 File). Excess hospital days for colon cancer cases were associated with stage and comorbidities (Fig 4; Table H in S1 File).

For rectal cancer cases, costs were higher for those with metastatic or regional disease, higher comorbidity score and diagnosis at age <65, with lower costs for cases with no health insurance, healthcare concession card holders and cases with unknown disease stage (Fig 3; Table G in S1 File). Excess hospital days were associated with stage, comorbidities, and health insurance status, along with more hospital days for current smokers (Fig 4; Table H in S1 File).

## Continuing care phase

In the continuing phase, the annual mean excess costs were $6,779 for colon cases and $8,336 for rectal cases. Hospital-based care accounted for around half of the costs for both cancer

**Table 2. Mean excess costs for colon and rectal cases diagnosed 2006–2013 in the 45 and Up Study, by phase of care, source of costs and disease stage.**

| | Initial phase | Continuing phase (per year) | Terminal phase |
|---|---|---|---|
| **Colon cases (n)** | **1012** | **905** | **445** |
| Mean excess cost per case | $50,434 | $6,779 | $74,952 |
| Hospital-based care (%) | 79% | 50% | 77% |
| MBS (%) | 12% | 20% | 8% |
| PBS (%) | 9% | 30% | 14% |
| Localised stage | $36,077 | $2,249 | $69,305 |
| Regional stage | $56,774 | $7,446 | $75,823 |
| Distant metastases | $79,437 | $26,374 | $81,183 |
| Unknown stage | $30,887 | $5,229 | $39,020 |
| **Rectal cases (n)** | **484** | **443** | **181** |
| Mean excess cost per case | $60,877 | $8,336 | $67,733 |
| Hospital-based care (%) | 75% | 45% | 73% |
| MBS (%) | 16% | 20% | 9% |
| PBS (%) | 8% | 35% | 17% |
| Localised stage | $49,071 | $3,556 | $68,479 |
| Regional stage | $73,157 | $8,663 | $66,078 |
| Distant metastases | $82,117 | $38,666 | $68,817 |
| Unknown stage | $34,354 | $1,047 | $67,084 |

MBS: Medicare Benefits Schedule; PBS: Pharmaceutical Benefits Scheme.

sites, with subsidised medicines in the PBS accounting for around one-third, and subsidised medical services in the MBS accounting for 20% (Table 2). The mean number of excess days in hospital was 2 days per year for both cancer sites (Table 3). Chemotherapy accounted for most of the anti-cancer treatment in this phase, and it was more common for cases with metastatic and regional disease.

For colon cancer cases, costs varied significantly by stage of disease and age at diagnosis, comorbidity score, and smoking and socioeconomic quintile at baseline (Table G & Fig B in S1 File). Excess costs for rectal cancer only varied significantly by stage of disease at diagnosis. For both sites, cases with distant metastases at diagnosis had higher excess costs than those with localised disease. Number of hospital days varied significantly by age at diagnosis,

**Table 3. Mean excess hospital days for colon and rectal cases diagnosed 2006–2013 in the 45 and Up Study, by phase of care and disease stage.**

| | Initial phase | Continuing phase (per year) | Terminal phase |
|---|---|---|---|
| **Colon cases (n)** | **1012** | **905** | **445** |
| Mean excess hospital days | 16.0 | 1.7 | 34.1 |
| Localised stage | 11.6 | 0.3 | 35.4 |
| Regional stage | 18.9 | 2.5 | 37.3 |
| Distant metastases | 22.3 | 3.0 | 33.0 |
| Unknown stage | 7.5 | 2.2 | 19.6 |
| **Rectal cases (n)** | **484** | **443** | **181** |
| Mean excess hospital days | 18.5 | 2.0 | 30.4 |
| Localised stage | 15.9 | 0.7 | 29.7 |
| Regional stage | 22.9 | 2.5 | 31.1 |
| Distant metastases | 20.3 | 8.1 | 30.5 |
| Unknown stage | 10.1 | 0.2 | 29.1 |

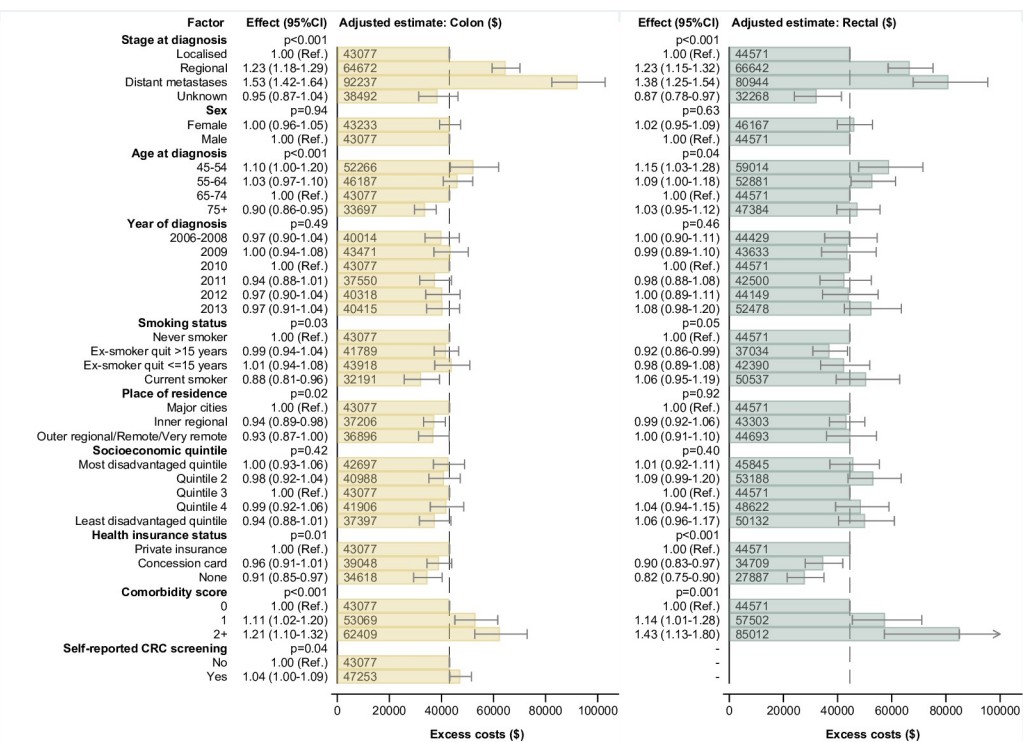

**Fig 3. Multivariable adjusted excess costs of colon and rectal cancer in the initial phase of care.** Notes: The dashed vertical line is the adjusted excess cost for a case in the reference category for all characteristics (i.e. $43,077 for colon cancer cases and $44,571 for rectal). For each category of a characteristic, the estimate shown reflects the adjusted excess cost for a case with all other characteristics in the reference category. To allow regression with non-negative values, the models were constructed using an offset of +$50,000; this offset was then deducted to obtain the adjusted estimates shown in the figure.

comorbidity score, and socioeconomic quintile at baseline for colon cases, and by stage of disease at diagnosis, comorbidity score, and smoking status for rectal cancer cases (Table H & Fig C in S1 File).

## Terminal care phase

We analysed the terminal care phase for 445 colon cancer cases and 181 rectal cancer cases who died during the study period. The mean excess costs were $74,952 for colon cases and $67,733 for rectal cases. Of cases who died, 66% had a record of dying in hospital. For both cancer sites, hospital-based care accounted for around three-quarters of the excess costs (Table 2). The mean numbers of excess days in hospital in this phase were the highest of all phases, with 34 days for colon cancer cases and 30 days for rectal cancer cases. There was little difference by disease stage, other than fewer days for colon cases with unknown stage (Table 3). The proportion of excess hospital days in public hospitals in this phase appeared to be higher than for other phases (Table J in S1 File).

After adjusting for all covariates, excess costs varied significantly by a few characteristics (Table G & Fig D in S1 File), and in particular were lower for both colon and rectal cases aged ≥75 at diagnosis compared to those aged 65–74. For colon cases, there were also lower excess costs for those with a healthcare concession card, those from rural or inner regional areas, and cases with unknown stage, while there were slightly higher costs for cases who self-reported having had an FOBT before baseline. However, days spent in hospital did not vary significantly

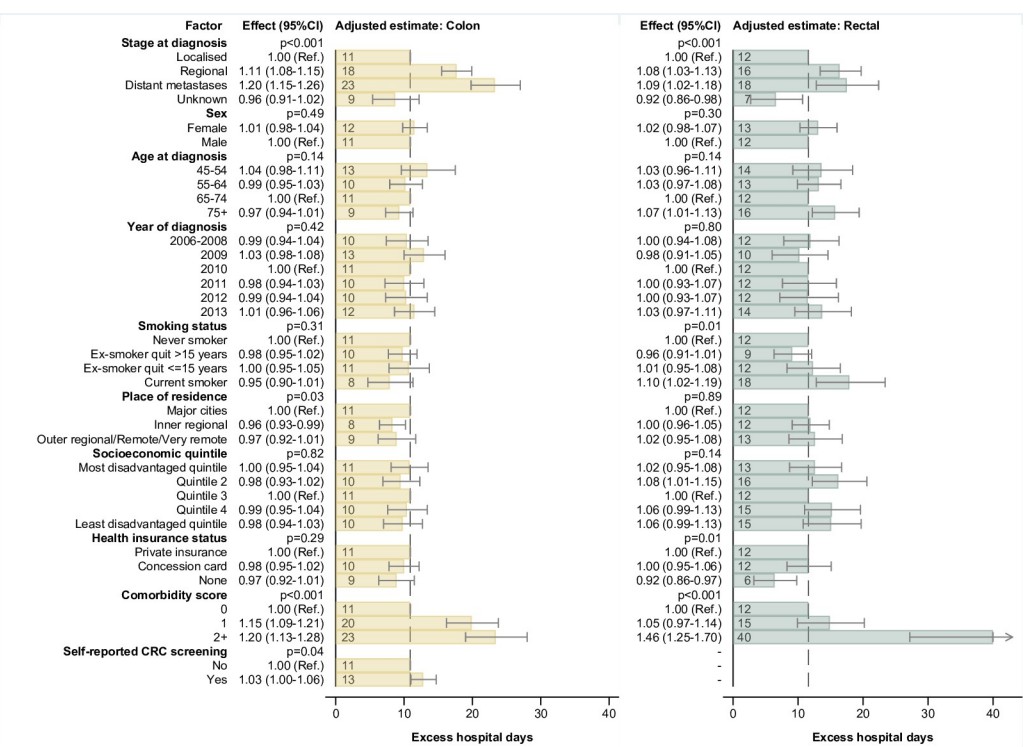

**Fig 4. Multivariable adjusted excess hospital days for colon and rectal cancer in the initial phase of care.** Notes: The dashed vertical line is the adjusted excess hospital days for a case in the reference category for all characteristics (i.e. 11 days for colon cancer and 12 days for rectal). For each category of a characteristic, the estimate shown reflects the adjusted excess days for a case with all other characteristics in the reference category. To allow regression with non-negative values, the models were constructed using an offset of +50 days; this offset was then deducted to obtain the adjusted estimates shown in the figure.

by most of the included covariates, with only some marginally significant associations (Table H & Fig E in S1 File).

## Annual/Monthly excess costs relative to diagnosis and death

The mean excess costs in the year prior to diagnosis were $3,338 for colon cancer cases ($2,113 in the month pre-diagnosis) and $2,453 for rectal cancer cases ($1,133 in the month pre-diagnosis), each with around one excess day in hospital in that year. In the year after diagnosis, the mean excess costs were $55,951 and $62,971 respectively, with a mean of 20 excess days in hospital. Colon cases had mean excess costs of $26,239 in the month after diagnosis compared with $16,728 for rectal cases, however rectal cases had consistently higher mean monthly costs from the second month onwards. There was a similar pattern for excess days in hospital (Tables I-L in S1 File). Among cases who died, the mean excess costs in the final month of life were ~$20,000, with hospital-based care accounting for >90%, and a mean of 11 excess days in hospital. ED presentations accounted for 6% of excess costs in the year and month pre-diagnosis, but in all other time periods and phases it accounted for <5% of excess costs (generally 2–3%).

## Discussion

We found substantial variation in costs and number of days in hospital by stage at diagnosis, highlighting cancer stage as a key factor in any analyses of CRC resource utilisation. These

results are particularly important for evaluating cost-effectiveness and resource utilisation for interventions that impact stage of disease, such as screening. This is the first comprehensive, population-based study in Australia to report CRC-related health system costs and days in hospital by phase of care, cancer site and disease stage at diagnosis.

A major advance in this study is that it provides detailed cost and hospitalisation information for key patient subgroups, using individual-level data from a population-based sample. To date, information reported in this way has been scarce, often due to a lack of available data. Previous studies in England, the US, and New Zealand have reported the excess health system costs of care for CRC cases relative to cancer-free controls by age and sex [19–21], and we previously reported costs in Australia for all CRC cases combined [5], but only the English study provided a breakdown of costs by cancer stage (Stages 1–2 versus 3–4) [20], and this is an area with little reported information. A previous Australian study in another state estimated stage-specific costs using a Victorian hospital-based sample from 2003–2010 –they reported similar trends of increasing costs with more advanced stage [14], as did the English study from 2001– 2010 [20]. We were able to extend the Australian information to a population-based sample and include a wider range of care for CRC patients (e.g. extra GP/specialist consultations), not limited to items flagged as specific to CRC or based on estimated case proportions likely to require certain care.

The impact of stage of disease at diagnosis was clearly apparent in the initial and continuing phases, where participants with metastatic disease at diagnosis had substantially higher healthcare costs and days in hospital than those with localised disease. Costs and number of days in hospital were also higher for participants with regional disease, although not as high as for those with metastatic disease. The small proportion of participants with "unknown" disease stage (recorded in the NSWCR) tended to have lower costs and fewer hospital days. Participants with "unknown" disease stage likely had less diagnostic work-up and less intensive treatment than other cases, as the classification of unknown stage indicates that insufficient information was available or provided to the cancer registry up to 4 months after diagnosis to assign stage (e.g. pathology from a surgical procedure), even if it was known to the treating clinician [30]. Having lower costs or fewer hospital days, particularly for cases with unknown stage, is not necessarily the optimal outcome. However, our finding that cases with earlier stage disease had lower costs in the initial and continuing care phases suggests that earlier detection, aside from having survival and quality of life benefits, could also have economic and hospital capacity benefits, although this could be partially countered by increased costs from an extended continuing phase. These detailed estimates by stage and phase will help estimate the health and economic benefits of early detection and screening.

Age at diagnosis was also an important determinant of resource utilisation. Compared with younger cases, colon and rectal cancer cases aged 75+ had lower costs, as has been reported previously [14, 19–21], and colon cancer cases aged 75+ had fewer hospital days. The reduced excess costs and hospital days for elderly participants in our study were not due to an increase in resources utilised by the matched controls as underlying age-related changes were accounted for by the age-matching process, and we found that the costs and numbers of days for controls increased only slightly compared with the large decline in actual costs and numbers of days for cases (relative to younger cases). The differences by age are most likely related to older participants having lower treatment rates and intensity, as has been reported by others [31]. We found that a substantially smaller proportion of participants aged ≥75 years received chemotherapy or rectal surgery than was observed for those aged <75, and older participants may have had less intensive follow-up care.

In the terminal care phase, there was little variation in costs or numbers of hospital days by any of the measured characteristics except age. The patterns of resource utilisation at the end

of life align with those described in a 2014 review, which reported most resource utilisation occurred at the very end of life, fewer resources for very elderly decedents, costs mainly incurred through hospitalisations, and no clear pattern by cancer type [32]. The estimated numbers of hospital days in the terminal phase also align with a study of all people who died aged 65+ in NSW during 2002–2003, with 29 hospital days during the final year for those who died from cancer (23 days for all causes) and decreasing numbers of days with increasing age [33].

Variations in resource utilisation were observed for other factors such as comorbidities, private health insurance, place of residence, and smoking, but they were generally not as large or consistent as those for stage and age. In the initial and continuing care phases, cases with more (or more serious) comorbidities had greater resource utilisation, most likely due to greater care requirements relating to the comorbid conditions. The variation was sometimes greater for rectal cancer cases, although this was based on a relatively small number of cases with a comorbidity score of 2+. Participants with colon cancer who did not live in a major city had lower costs than those in major cities during the initial and terminal care phases, perhaps due to treatment accessibility barriers limiting the amount of hospital-based or subsidised follow-up care, especially for patients who needed to travel long distances and/or find accommodation away from home. Compared with cases with private health insurance, cases without private health insurance had lower costs in the initial phase, and those with colon cancer had lower costs in the terminal phase. This might be due to private health insurance being a proxy for health literacy or health-seeking behaviour, which can lead to people having more expensive or additional treatments. The differences in costs by health insurance could also be an artefact of how the costs were estimated, with potential for double-counting of some services in private hospitals that are also recorded in the MBS. After excluding all hospital-based MBS records, costs declined by ~5% across the three phases and most of the cost differences by insurance status were not statistically significant, while there were only minimal changes in the associations between costs and all other measured characteristics.

Among current smokers there were mixed results for the initial phase: rectal cancer cases who were current smokers had more hospital days than non-smokers, and colon cancer cases who were current smokers had lower costs than non-smokers. These patterns could be due to smokers having other health conditions that require hospital-based care but limit more intensive cancer treatment–in our sample smokers appeared to be less likely to have surgery and more likely to have chemotherapy. Among rectal cancer cases, ex-smokers who quit >15 years earlier had fewer hospital days during the initial phase than current smokers (Fig 4; Table H in S1 File). This may indicate that the health benefits from quitting smoking also help reduce the need for hospital-based care. There were no strong associations between BMI and costs of care, although there was a suggestion of fewer hospital days during the terminal phase for rectal cancer cases with very high BMI. Any actual association with BMI may have been attenuated due to potential measurement error from self-reported height and weight, and by changes in BMI between baseline and cancer diagnosis or treatment.

Year of diagnosis was not associated with changes in CRC care costs across the period 2006–2013. The previous Australian hospital-based CRC costing study highlighted the impact of specific new drugs on treatment costs for advanced disease in 2011, estimating an additional AUD$10,000–12,000 for each metastatic case using bevacizumab or cetuximab [14]. The use of new targeted therapies is expensive at the individual patient level, but our results suggest that any changes to CRC care during 2006–2013 did not substantially impact overall health system costs (population level), however others have reported increasing subsidised drug costs for all cancers combined during that time [34]. More recent data are required to assess the impact of an ever-increasing range of new technologies, targeted therapies, and indications for

CRC detection and treatment. Also, year of diagnosis was not associated with the number of days in hospital during the study period. An English study reported a decline in the median length of hospital stay for CRC surgery admissions, from 10 days in 1998 to 7 days in 2010 [35]–we found a consistent median of 9 days per surgery admission over the period 2006–2013, which is consistent with the 8 days reported in a 2019 audit from the CRC Surgical Society of Australia and New Zealand [36], and suggests that there were no major changes in surgical practice or changes in complication rates in NSW during this period.

We could not evaluate differences in costs for cancers detected through screening compared with others, as we did not have data on whether participants were diagnosed via screening. The self-reported FOBT/screening information was for activity prior to study baseline, and therefore is unlikely to refer to any screening episodes that led to the diagnosis of CRC. A previous analysis of the 45 and Up Study demonstrated that participants who self-reported screening at baseline were 44% less likely to be diagnosed with CRC within 4 years of follow-up [37]. However, the NBCSP was in the early stages of its phased implementation during the period when data accrued for the current analysis, and ad hoc, community-based FOBT screening was minimal, so any impact of screening on the overall cost of CRC across the study period was likely to be minimal. We hypothesise that CRC management would cost less and require less hospital time per case if CRC screening, via the NBCSP, had been fully rolled out during the period for which data were accrued, since effective screening is known to change the stage distribution in the population towards an earlier stage at diagnosis [38].

Importantly, the results reported here can be used to inform health economic evaluations of screening and other interventions targeted at reducing the burden of CRC. For example, these results will be used in our team's comprehensive microsimulation modelling platform that is being developed for CRC in Australia [39].

This study has some limitations. We did not include all possible resource-related information, with missing information on some non-admitted hospital costs (e.g. community-based care), treatment that may have been supplied by access programs [40], and numbers of hospital days did not include outpatient (non-admitted) procedures. We focused on reporting means of excess costs, which can be influenced by extreme (high/low) values. However, the large sample size makes the estimates more robust to extreme outliers, limiting their influence, as does the use of multiple matched controls for each case. We have also reported other measures such as medians and standard deviations. Some groups of cases will have influence over the mean values, but this represents actual resource use and it is important to include these in estimates.

The 45 and Up Study had a response rate of 18%, so the cohort might not be representative of the NSW population, however this should be offset somewhat by using matched controls from within the cohort. Additionally, the study sample was potentially not representative of all people diagnosed with CRC in NSW. For both cancer sites, 6% of all people in NSW with CRC were aged <45 years at diagnosis and participants aged <45 years were not eligible to participate in the 45 and Up Study. As there was little variation in resource use for participants aged under 75 years, the impact of this exclusion may be limited. There was also an over-representation of people with private health insurance, and people from regional and remote areas in the study sample due to the 45 and Up Study's sampling strategy, potentially underestimating some healthcare utilisation for colon cancers where those in major cities had higher resource utilisation. Any potential non-representativeness of the sample may have an impact on some of the estimated absolute costs or numbers of hospital days. However previous work has shown that health services utilisation for CRC cases in this cohort is reasonably representative of all CRC cases in NSW [41], suggesting limited impact of any non-representativeness on the estimates from this study.

Cases were matched to controls by age, sex, geography, and smoking status to estimate excess resource utilisation due to colorectal cancer. This extends the age-sex matching used in prior studies [19–21], but we did not explicitly control for other potential confounders through matching. There was little association between costs/days and BMI or socioeconomic level, so additionally matching on these factors would have little impact. The main associations with costs/days were for stage (not applicable to controls), age (a matching variable) and comorbidities. There were slightly more comorbidities recorded for cases than controls, but these mainly appeared or were recorded at hospital admissions in the 6 months leading up to diagnosis, and cases and controls had very similar self-reported health at baseline (Table C in S1 File). We did not include physical activity, alcohol consumption, or diet in the analysis, as these were beyond the scope of this study.

It is important to note that our study focused on direct healthcare costs incurred after a cancer diagnosis. We did not include patients' out-of-pocket costs or indirect/societal costs, which are also key components of the entire burden of a cancer diagnosis on the community and we will address them in future work. We also did not have detailed information on participants' comorbidities, only using conditions recorded during hospitalisations, nor did we have information on patients' quality of life or treatment decisions–all of which would give a more complete picture of the healthcare experience for CRC patients. Also, the staging information available (localised, regional, distant metastases, unknown) does not include the detail provided by the TNM and other staging systems used in treatment guidelines [4], but it does give a meaningful and useful level of detail to differentiate between cancer patients that can be used in future modelling.

However, this study has several key strengths. We used detailed person-level information from a large prospective cohort study, and with the use of administrative health records, the resource utilisation data were not affected by recall bias, and provided comprehensive coverage of health services used [42, 43]. We also used self-reported information, such as smoking status and BMI at baseline, providing personal information that is difficult or impossible to obtain from routinely collected administrative data alone. We were able to capture a wide range of healthcare utilisation for a population-based cohort, improving estimates which previously had been based only on specific items directly attributable to CRC or predictions from hospital-based samples. We matched cases with multiple controls to give more robust results, with the size of the cohort and depth of available data allowing for matching of controls by several key characteristics, and cases were identified through a comprehensive population-wide registry. The results from this analysis will inform more detailed evaluations of the impact of CRC control interventions through costs and hospital requirements, and extend existing evaluations [9–13] to reinforce the benefit of interventions through changes in hospital utilisation. This will help identify the best use of future healthcare resources and determine cost-effective strategies to reduce the CRC burden.

## Conclusions

Health system costs and hospital utilisation for CRC are strongly associated with disease stage at diagnosis, often with increased resource requirements for people with more advanced disease. We have provided detailed data to estimate the variation in healthcare costs and hospital utilisation by stage within each phase of care, and highlighted other factors associated with resource utilisation. This study is timely in the Australian setting, with the full implementation of the national screening program and increasing availability of candidate targeted therapies and immunotherapies for the treatment of CRC.

## Supporting information

**S1 File. Detailed descriptive statistics, results, and figures.**
(DOCX)

## Acknowledgments

This research was completed using data collected through the 45 and Up Study (www.saxinstitute.org.au). The 45 and Up Study is managed by the Sax Institute in collaboration with major partner Cancer Council NSW; and partners: the Heart Foundation; NSW Ministry of Health; NSW Department of Communities and Justice; and Australian Red Cross Lifeblood. We thank the many thousands of people participating in the 45 and Up Study, the Centre for Health Record Linkage for the record linkage and Services Australia, the NSW Ministry of Health and Cancer Institute NSW for the use of their data. We thank Clare Kahn for editorial assistance.

## Author Contributions

**Conceptualization:** David E. Goldsbury, Eleonora Feletto, Marianne F. Weber, Dianne L. O'Connell, Karen Canfell.

**Data curation:** David E. Goldsbury.

**Formal analysis:** David E. Goldsbury.

**Methodology:** David E. Goldsbury, Eleonora Feletto, Marianne F. Weber, Philip Haywood, Alison Pearce, Jie-Bin Lew, Joachim Worthington, Emily He, Julia Steinberg, Dianne L. O'Connell, Karen Canfell.

**Writing – original draft:** David E. Goldsbury.

**Writing – review & editing:** David E. Goldsbury, Eleonora Feletto, Marianne F. Weber, Philip Haywood, Alison Pearce, Jie-Bin Lew, Joachim Worthington, Emily He, Julia Steinberg, Dianne L. O'Connell, Karen Canfell.

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
