## [Decision Letter · Decision Letter 0]

7 Sep 2021

PONE-D-21-14030Health system costs and days in hospital for colorectal cancer patients in New South Wales, AustraliaPLOS ONE

Dear Dr. Goldsbury,

Thank you for submitting your manuscript to PLOS ONE. After careful consideration, we feel that it has merit but does not fully meet PLOS ONE’s publication criteria as it currently stands. Therefore, we invite you to submit a revised version of the manuscript that addresses the points raised during the review process.

Two reviews are attached. Reviewer 1 makes an important point (Comment 2) about the several analyses and tables presented. Responding to this clearly may also address some of Reviewer 2's comments. While reviewer 2 has several suggested changes, I believe many of these can be easily addressed though adding more detail for clarity. 

We look forward to receiving your revised manuscript.

Kind regards,

Anna Ugalde, PhD

Academic Editor

PLOS ONE

Journal Requirements:

Additional Editor Comments (if provided):

Thank you for the submission to Plos One, and following on from our email correspondence, please accept my apologies for the delay in securing peer reviewers for this manuscript.

Two reviews have been conducted, both are supportive of publication of this work. One review, by Reviewer 2, has identified a few areas that require slight changes in order to be clearer for the reader. Reviewer 1 requests: 'There are several analyses / tables presented, and I wondered about the rationale for looking at costs and hospital days in each of these ways – the 3-phase approach, the yearly 2-years pre-diagnosis to 5-years post-diagnosis data, as well as 3-month pre-diagnosis to 6-months post-diagnosis?', I think a clear statement about the rationale addressing this may also address some of reviewer 2's concerns and queries about the analytic approach.

I hope these reviews are helpful in refining your paper.

Reviewers' comments:

Reviewer's Responses to Questions

**Comments to the Author**

1. Is the manuscript technically sound, and do the data support the conclusions?

Reviewer #1: Yes

Reviewer #2: Yes

2. Has the statistical analysis been performed appropriately and rigorously? 

Reviewer #1: Yes

Reviewer #2: Yes

3. Have the authors made all data underlying the findings in their manuscript fully available?

Reviewer #1: Yes

Reviewer #2: No

4. Is the manuscript presented in an intelligible fashion and written in standard English?

Reviewer #1: Yes

Reviewer #2: Yes

5. Review Comments to the Author

Reviewer #1: Thank you for the opportunity to review this excellent paper. The study uses a case-control design to investigate mean excess costs and days in hospital associated with a colon or rectal cancer diagnosis. The writing is very clear, methodology sound, and there are some interesting findings. The work is presented mainly as a preliminary paper to assist later modelling studies of cancer screening/prevention or treatment initiatives.

There are several analyses / tables presented, and I wondered about the rationale for looking at costs and hospital days in each of these ways – the 3-phase approach, the yearly 2-years pre-diagnosis to 5-years post-diagnosis data, as well as 3-month pre-diagnosis to 6-months post-diagnosis?

For an international audience, it would be good to clarify briefly what kind of services (relevant to CRC) are provided by data from MBS, e.g. should note this includes GP attendance.

The rationale for choosing socio-demographic/clinical factors associated with cost/days was not very well articulated. Results were then a little difficult to determine what was expected vs unexpected findings.

It was helpful to see data not just on mean costs/hospital days, but also median/IQR and SD. Shows some pretty substantial variation, including into negative costs/days at some timeframes, vs controls (Table C) and suggests some cases – possibly a relatively small number – influencing costs. If there is room, it could be helpful to comment on this as it may be relevant for service providers to consider when organising CRC care.

Interesting findings regarding insurance status – it is suggested this could be due to different behaviours of people who have insurance, though services use might be related to how care is provided by public/private care providers.

Reviewer #2: Thank you for the opportunity to review this well considered manuscript. This will undoubtedly contribute to evaluations of screening and CRC management in Australia in the coming years.

The matched controls are described as

'matched by age, sex, geography, smoking' whilst the latter two variables offer some measure of correction for risk factors please clarify any precedents for this approach and acknowledge any potential shortcomings i.e. the limitation of obesity, alcohol consumption, or other SES related factors as known risk factors for CRC - This is pertinent since the subsequent stratification uses obesity as a classifier.

Line 165 onward

'This left 1% of admissions without an AR-DRG – almost all >2 years after diagnosis – and these were assigned a cost of $1500 for the first day (compared with single-day costs of ~$2200 for a colonoscopy and ~$1000 for rehabilitation) and $300 per additional day. A sensitivity test using double these assigned costs made <1% difference to the results.'

This comes across a little unclear? What do you mean? Is it that an alternative A or B costing was compared ... or that speculatively that a colonoscopy would be the rationale for any admission during this time? if so why ? Surely after a defined time frame it would not be an accepted standard that just because someone had a CRC that a scope would be indicated during an admission... surely a CT might be the first test of preference clinically?

Having read on it appeared this was a comparison ... therefore please frame the intro to the preceding sentence with the word 'either' and consider providing a rationale as to why a colonoscopy is the default test of preference at such a time point - in the absence of specific GI symptoms it would be more clinically likely that a CT would be ordered first as an inpatient if Hx of CRC were noted and the presenting condition thought in any way connected - ie they would want to exclude mets, rather than per se to re-examine the bowel itself +/- the chances that the individual had a rectal tumour and stoma placement etc...

Statistical analysis line 197

It might be useful if you had a small diagram to show this

eg

Case -----------------

C1 --------X

C2 ----------------X

C3 ----------------

or something similar

Case -----X

C1 -----X------------

C2 -----X------------

C3 -----X------------

Line 207

'Hospitalisations could span multiple time periods, so costs were apportioned across time periods using the proportion of days in each time period out of the total length of stay'

This is not super clear what that means... ? as in if a patient had a 3 week inpt stay which crossed monthly cut points? or straddled a given year end? It would be helpful to clarify (based on the assumption that this work may form the basis for future updates of the work.)

Line 211

How does the defined staging system compare to TNM Stage 1-4 ? and will that be useful for future work where the treatment indications are not by this staging system? Was TNM not available?

Table 1

I would assume the layout will differ in print... but having the title layout not cross / doesn't hang over multiple pages makes it hard to read (at 6pm on a Friday...)

RE Private insurance coverage - How would this compare to the general Australian population? is that an expected level of private coverage.

4th paragraph in the Discussion on age (last sentence)

'A substantially smaller proportion of participants aged ≥75 years received chemotherapy or rectal surgery than was observed for those aged <75, and older participants may have had less diagnostic work-up'

How is this statement supported by the data... are they more represented in the stage unknown? just confused here, one can't be both staged and then have less diagnostic work up?

They would have less surgical assessments, less ongoing CT monitoring, less ongoing blood tests ... follow up of care still in the time frame of your categorization of initial care ... but they would still have similar diagnostics to determine stage?

6th paragraph

'Compared with current smokers, ex-smokers who quit >15 years earlier had fewer hospital days during the initial and continuing phases. ' Please provide an effect size, or direct the reader to the section in the supplementary information.

In limitations you state

'We did not include all possible resource-related information, with missing information on some non-admitted hospital costs (e.g. community-based care), treatment that may have been supplied by access programs (38), and numbers of hospital days did not include outpatient (non-admitted) procedures'

Have you clarified if colonoscopy is actually an outpatient procedure (as would be standard in the UK) - if so the attribution of a scope above and the $2200 shortfall should be acknowledged in the use of these costs for future researchers modelling as you suggest - clarification here would therefore be essential

2nd last paragraph

'important to note that our study focused on direct healthcare costs incurred after a cancer diagnosis. '.....

Why then in the discussion did you differentiate / suggest that diagnostic work up in older patients was likely… you couldn’t therefore determine that

Perhaps I wasn’t fully alert above – but a clear statement in the methods should indicate that no diagnostic costs are included – or at least that all costs are incurred after diagnosis.

Conclusions

'The results from this analysis will inform more detailed evaluations of the impact of CRC control interventions through costs and hospital requirements, and extend existing evaluations (9-13) to reinforce the benefit of interventions through changes in hospital utilization.'

Why are there references in the conclusion? Surely that’s for the discussion… but also you’re preempting the results of future work that hasn’t been done here…

Focus on your own conclusions…

'This will help identify the best use of future healthcare resources and determine cost-effective strategies to reduce the CRC burden. '

This is not a direct conclusion of this work, amend the previous sentence to consider some of the framing you wish… but otherwise move into the discussion.

Supplementary Info reviewed - tracked comments enclosed - minor clarifications.

6. PLOS authors have the option to publish the peer review history of their article (what does this mean?). If published, this will include your full peer review and any attached files.

Reviewer #1: No

Reviewer #2: No

---

## [Editor Report · Decision Letter 1]

3 Nov 2021

Health system costs and days in hospital for colorectal cancer patients in New South Wales, Australia

PONE-D-21-14030R1

Dear Dr. Goldsbury,

We’re pleased to inform you that your manuscript has been judged scientifically suitable for publication and will be formally accepted for publication once it meets all outstanding technical requirements.

Kind regards,

Anna Ugalde, PhD

Academic Editor

PLOS ONE

Thank you for responding to the reviewers comments, I think this manuscript will be an important contribution to the journal and our understanding of hospital costs for colorectal cancer patients.
---

## [Editor Report · Acceptance letter]

17 Nov 2021

PONE-D-21-14030R1 

Health system costs and days in hospital for colorectal cancer patients in New South Wales, Australia 

Dear Dr. Goldsbury:

I'm pleased to inform you that your manuscript has been deemed suitable for publication in PLOS ONE. Congratulations! Your manuscript is now with our production department. 

Kind regards, 

on behalf of

Dr. Anna Ugalde 

Academic Editor

PLOS ONE